# End-to-end Quantized Training via Log-Barrier Extensions

## Abstract

Quantization of neural network parameters and activations has emerged as a successful approach to reducing model size and inference time on hardware that supports native low-precision arithmetic. Fully quantized training would facilitate further computational speed-ups as well as enable model training on embedded devices, a feature that would alleviate privacy concerns resulting from the transfer of sensitive data and models that is necessitated by off-device training. Existing approaches to quantization-aware training (QAT) perform "fake" quantization in the forward pass in order to learn model parameters that will perform well when quantized, but rely on higher precision variables to avoid overflow in large matrix multiplications, which is unsuitable for training on fully low-precision (e.g. 8-bit) hardware. To enable fully end-to-end quantized training, we propose Log Barrier Tail-bounded Quantization (LogBTQ). LogBTQ introduces a loss term, inspired by the log-barrier for constrained optimization, that enforces soft constraints on the range of values that model parameters can take on. By constraining and sparsifying model parameters, activations and inputs, our approach eliminates overflow in practice, allowing for fully quantized 8-bit training of deep neural network models. We show that models trained using our approach achieve results competitive with state-of-the-art full-precision networks on the MNIST, CIFAR-10 and ImageNet classification benchmarks.

## 1 Introduction

As state-of-the-art deep learning models for vision, language understanding and speech grow increasingly large and computationally burdensome (He et al., 2017; Devlin et al., 2018; Karita et al., 2019), there is increasing antithetical demand, motivated by latency, security and privacy concerns, to perform training and inference in these models on smaller devices at the edge rather than in server farms in the cloud. Model quantization has emerged as a promising approach to enable deployment of deep learning models on edge devices that reduce energy, latency and storage requirements by performing floating-point computation in low precision (less than 32 bits).

There are two primary strategies for quantization: *Post-training* approaches quantize the parameters of a model trained in full precision post-hoc, and tend to suffer a heavy penalty on accuracy since their inference graph differs substantially from training (Jacob et al., 2018). *Quantization-aware training* (QAT) (Bhuwalka et al., 2020) combats this discrepancy by simulating quantization during training, so that model parameters are learned that will work well when inference is performed in low precision. In this work, we focus on the latter setting, suitable for fully quantized training on low-precision (e.g. 8-bit) devices.

Though QAT results in quantized models that perform largely on par with their non-quantized counterparts, current state-of-the-art QAT methods (Wu et al., 2018; Wang et al., 2018; Bhuwalka et al., 2020) are not suitable for training on fully low-precision hardware because they employ *fake quantization*, meaning each operation is executed using 32- or 16-bit floating point arithmetic, and its output is quantized to lower precision, e.g. int8. This results in two key incompatibilities with fully low-precision training, and consequently deployment on real low-precision hardware. First, existing QAT approaches assume perfect sums in inner product operations, which means that the accumulators used to compute matrix multiplies (the **acc** row in Table 1) must be higher precision than the values being multiplied (other bit-precision rows in Table 1). This is to avoid losing res-

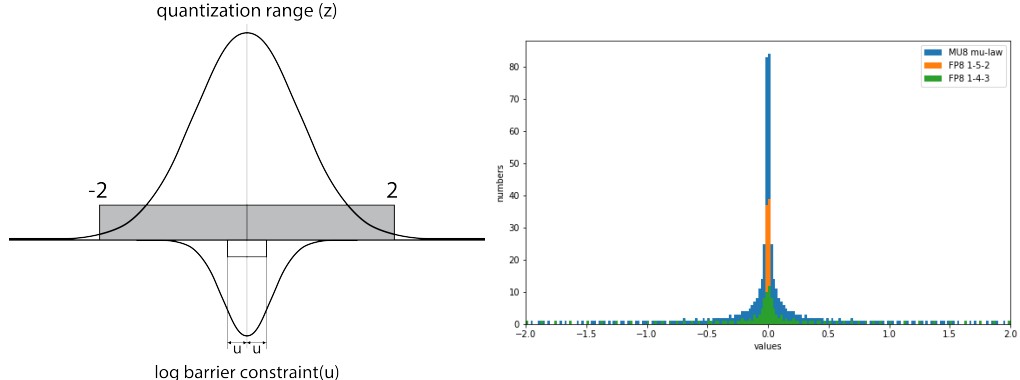

Figure 1: Left: Visualization of the log barrier constraint applied to network parameters quantized in range $[-2, 2]$. See §3.3 for an approximated tail bound on possible overflow. Right: $\mu$-law encoding vs. FP8(1-5-2) and FP8(1-4-3) for all possible values on interval $[-2, 2]$. $\mu$-law maintains higher precision at concentrated small values.

olution in low-precision additions, also known as *swamping* (Wang et al., 2018)[1]. Second, QAT commonly leverages dynamic quantization ranges per-layer, meaning the mapping between high- and low-precision values varies by layer, carefully tuned as a function of the network architecture, optimization dynamics and data during training. While this practice results in higher quantized inference accuracy, it is also a challenge to low-precision training, since it is unclear how to tune those ranges when training on new data in the absence of high-precision arithmetic. These incompatibilities present a substantial hurdle to quantized training in practice. For example, an automotive electronics manufacturer may want to deploy a machine learning model on its 8-bit door lock or power window controller to adaptively fit the users' habits. In this scenario, existing approaches for quantized training would fail (Sakr et al., 2019).

In response, we propose a new approach for fully quantized training of neural network models, inspired by the *barrier method* from convex optimization (Boyd & Vandenberghe, 2004). *Log Barrier Tail-bounded Quantization* (LogBTQ) utilizes a log barrier extension loss (Kervadec et al., 2019) to constrain the output of the network, encouraging all model parameters and activations to stay within the same predefined range. The log barrier function itself is a smooth approximation of the indicator function, which is ideal for selecting the weights that are within the range of quantization (see Figure 1, left). By fixing a single quantization range throughout the network at the beginning of training, our approach both obviates the need for dynamic ranges, and the limits of the range are set so as to alleviate overflow[2] in matrix multiply accumulations. We combine the log barrier extension loss with an $L^1$ regularization term (Hoffer et al., 2018) to further reduce the total magnitude of parameters and activations in the model. To allow for gradients, which tend form a peaky distribution near extremely small values (Zhou et al., 2016; Jain et al., 2020), to be quantized using the same range as the rest of the network, we also adopt the nonlinear $\mu$-law algorithm from audio applications (Deng & Doroslovacki, 2006) to construct a new MU8 codebook that better deals with "swamping" issues compared to the standard IEEE Float Standard. Experiments show that our approach achieves competitive results compared to state-of-art full-precision models on the MNIST, CIFAR-10 and ImageNet classification benchmarks, despite our models being trained end-to-end using only 8 bits of precision.

---

[1]Swamping: Accumulation of floating-point numbers, where the small magnitude value is ignored (or truncated) when it is added to the large magnitude sum.

[2]Overflowing: for the fixed-point accumulation where the accumulated value wraps around to the small value when it exceeds the largest value representable by the given accumulation precision.

| Quantized Training Scheme | Training Precision | | | | | Quant. Range | Train/Test Prec. | |
|---|---|---|---|---|---|---|---|---|
| | W | x | dW | dx | acc | | FP32 | Low |
| DoReFa-Net (Zhou et al., 2016) | 1 | 2 | 32 | 6 | 32 | Dynamic | *55.9* | *46.1* |
| WAGE (Wu et al., 2018) | 2 | 8 | 8 | 8 | 32 | Dynamic | — | *51.6* |
| DFP (Das et al., 2018) | 16 | 16 | 16 | 16 | 32 | Dynamic | 75.70 | 75.77 |
| MPT (Micikevicius et al., 2018) | 16 | 16 | 16 | 16 | 32 | Dynamic | 75.92 | 76.04 |
| Wang et al. (2018) | 8 | 8 | 8 | 8 | 16 | Dynamic | 72.14 | 71.72 |
| HFP8 (Sun et al., 2019) | 8 | 8 | 8 | 8 | 8/16 | Dynamic | 76.44 | 76.22 |
| **LogBTQ (ours)** | **8** | **8** | **8** | **8** | **8** | **Fixed** | 73.59 | 71.11 |

Table 1: Comparison of reduced-precision training for top-1 accuracy (%) using ResNet-50 (ImageNet). For works that did not evaluate on ResNet-50, we include AlexNet results (italicized). *Dynamic* indicates that quantization ranges vary by layer and must be learned or tuned; *Fixed* indicates a single quantization range is fixed globally throughout the network.

## 2 BACKGROUND AND RELATED WORK

### 2.1 POST-TRAINING QUANTIZATION

There was been a recent surge of interest in quantization research. In 2020 alone, there were a number of important developments in post-training quantization. Rusci et al. (2020); Jain et al. (2020); Esser et al. (2020); Uhlich et al. (2020) proposed learning-based approaches for determining the quantization ranges of activation and weights at low precision. Stock et al. (2020) advocates preserving the quality of the reconstruction of the network outputs rather than its weights. They all show excellent performance compared to full-precision models after quantization. Sakr & Shanbhag (2019) presented a detailed analysis of reduced precision training for a feedforward network that accounts for both the forward and backward passes, demonstrating that precision can be greatly reduced throughout the network computations while largely preserving training quality. Our work share the same intuition of preferring small predetermined dynamic range (PDR) and small clipping rate[3]. However, Sakr & Shanbhag (2019)'s approach requires the network first be trained to convergence at full 32 bit precision, which is a significant limitation. In this paper, we focus on training rather than inference on low-precision hardware, therefore, we do not assume access to a full-precision high-performing model as a starting point.

### 2.2 QUANTIZATION-AWARE TRAINING

Pioneering works in this domain (Zhou et al., 2016; Courbariaux et al., 2015) looked at quantizing model weights, activations, gradients to lower precision to accelerate neural network training. The terminology *quantization aware training* (QAT) was first introduced by Jacob et al. (2018). QAT incoporates quantization error as noise during training and as part of the overall loss, which the optimization algorithm tries to minimize. Hence, the model learns parameters that are more robust to quantization, but QAT is not meant to be performed entirely in low precision, it aims to learn parameters that will work well for low-precision inference. More recently, several works further pursued the goal of enabling fully low-precision training (Wu et al., 2018; Wang et al., 2018; Das et al., 2018; Sun et al., 2019). As shown in Table 1, most existing work employs *fake* quantization, resorting to higher precision values to compensate for the swamping issue, especially during gradient accumulation. Mixed-precision quantization (Das et al., 2018; Wang et al., 2018; Zhang et al., 2020a), which quantizes a neural network using multiple bit precisions across layers, still relies on higher-precision gradients to preserve model accuracy. This means it is difficult, if not impossible, to implement these approaches on low-bit (e.g. 8-bit) hardware.

Most similar to our work, Sun et al. (2019) claim it is possible to do every step in low precision, but the quantization range for the layers in their work is very carefully chosen empirically, which presents great difficulty if we were to train models from scratch on low-precision hardware. Their method also requires a copy of the quantization error (residual) in FP16(1-6-9) (hence 8/16 in Ta-

---

[3]see Appendix B of Sakr & Shanbhag (2019) explaining PDR; refer to their Criterion 2 about clipping rate.

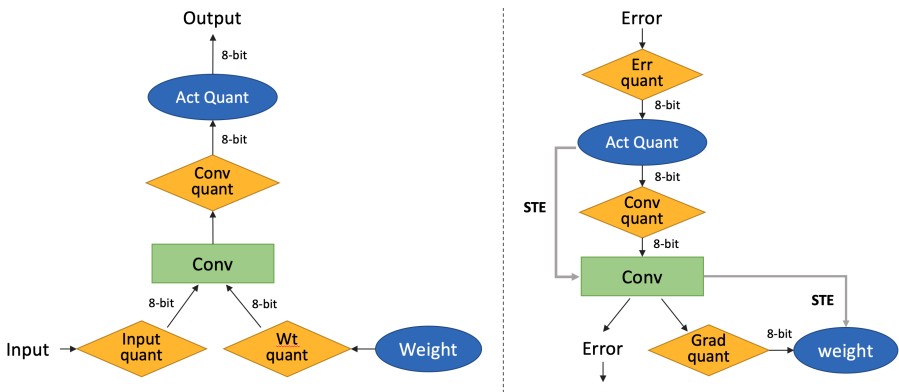

Figure 2: Left: Diagram of the forward pass using LogBTQ quantization. Right: Gradient propagation.

ble 1). In addition to the 9-bit mantissa, the exponent bit in their floating point format would need to be manually modified to store the residual due to its small value.

In this paper, we propose a new quantization scheme: log-barrier tail-bounded quantization (Log-BTQ) that can perform fully end-to-end low precision training, suitable for deployment on low-precision hardware. Our major contributions are the following:

1. We apply a log barrier extension loss to soft-threshold the values of network weights and activations to constrain all the values to be small. Our quantization scheme also enables global fixed-range quantization which together significantly alleviates the overflow issue caused by large numbers and dynamic range.

2. We add an $L^1$ loss term to encourage sparsity and further reduce overflow.

3. We propose $\mu$-law quantization (MU8) instead of INT8, FP8(1-4-3) or FP8(1-5-2) to construct a more accurate codebook that better compensates for the peaky concentration of network parameters around small values.

## 3 LOG BARRIER TAIL-BOUNDED QUANTIZATION (LOGBTQ)

The overall diagram of our quantization scheme is shown in Figure 2 (left). Figure 2 (right) shows the backward pass, where we quantize everything at each layer and all operations including input $x$, weights $w$, activations $a$, errors $e$, and gradients $g$ (including the gradient accumulation step). We denote all these values as the set $Z = \{x, w, a, e, g\}$. In this work, different from previous works (Sakr & Shanbhag, 2019; Zhang et al., 2020b) that used adaptive quantization range, we adopt a globally fixed quantization range for every element $z \in Z$, and set $z \in [-2, 2]$. We do not need to adjust the range and precision during training as in other quantization work that relies on layer-wise dynamic ranges. This would greatly reduce the overhead for implementation on hardware.

### 3.1 CONSTRAINED FORMULATION

Let $\mathcal{D} = \{I^1, ... I^N\}$ denote the labeled set of $N$ training images, and $f$ denote the neural network model, $\theta$ here denotes all the parameters of the neural network including weights $w$. For task, such as image classification, we are usually solving such an optimization problem: $\min_{\theta} \mathcal{L}(f_\theta(I))$ where $\mathcal{L}$ is the loss function of our neural network training objective. In this work, we use the typical cross-entropy loss, and since we are interested in constraining the quantization threshold, we are effectively performing constrained optimization in such a form:

$$
\begin{aligned}
& \underset{\theta}{\text{minimize}} && \mathcal{L}\left(f_\theta(I)\right) \\
& \text{subject to} && |\theta_n| \leq u, \ n = 1, \ldots, N.
\end{aligned}
\tag{1}
$$

With $u$ our desired barrier (perturbation). In practice, we set $u = 0.1$ to ensure we can represent as much information as possible within our quantization range (Figure 1, left). This setting is also explained further in Section 3.3.

## 3.2 LOG-BARRIER EXTENSION FUNCTION

Theoretically, problem (1) should be best solved by the log barrier method which is an interior point method that perfectly handles inequality constraints. (Tibshirani, 2019; Boyd & Vandenberghe, 2004): In phase I, we would perform Lagrangian-dual optimization to find the feasible points:

$$\underset{\lambda}{\text{maximize}} \ \underset{\theta}{\text{minimize}} \quad \mathbb{L}(x, \lambda) = \mathcal{L}(f_\theta(I)) + \sum_{n=1}^{N} \lambda^n(|\theta_n| - u)) \tag{2}$$

$$\text{subject to} \qquad \lambda \succeq 0, \ n = 1, \dots, N.$$

where $\lambda \in \mathbb{R}_+^{1 \times N}$ is the Lagrangian multiplier (dual variable). After we find a feasible set of network parameters, we can use the barrier method to solve equation (1) as an unconstrained problem:

$$\underset{\theta}{\text{minimize}} \quad \mathcal{L}(f_\theta(I)) + \sum_{n=1}^{N} \psi_t(|\theta_n| - u) \tag{3}$$

solving problem (3) is Phase II, where $\psi_t$ is the standard log-barrier function: $\psi_t = -\frac{1}{t} log(-z)$. As $t$ approaches infinity, the approximation becomes closer to the indicator function. Also, for any value of $t$, if any of the constraints is violated, the value of the barrier approaches infinity. However, a huge limitation in applicability to practical problems such as ours is that the domain of Eq. (3) must be the set of feasible points. The canonical barrier method above is also prohibitively computationally expensive given there are millions of parameters in the network, and we need to alternate the training between primal and dual and do projected gradient ascent for the dual variable.

We are not particularly concerned with the weak duality gap to lower-bound the optimal solution in this work, instead, we are interested in the property of the barrier method to handle inequality constraints. Therefore, inspired by Kervadec et al. (2019), we formulate quantization as an unconstrained loss to approximate the constrained optimization problem:

$$\underset{\theta}{\text{minimize}} \quad \mathcal{L}(f_\theta(I)) + \sum_{n=1}^{N} \tilde{\psi}_t(|\theta_n| - u) \tag{4}$$

where $\tilde{\psi}_t$ is the log-barrier extension, which is convex, continuous, and twice-differentiable:

$$\tilde{\psi}_t(z) = \begin{cases} -\frac{1}{t} log(-z) & z \leq -\frac{1}{t^2} \\ tz - \frac{1}{t} log(\frac{1}{t^2}) + \frac{1}{t} & \text{otherwise} \end{cases} \tag{5}$$

in our case, the input $z$ to the log-barrier extension is the same $z$ we defined in the beginning of Section 3, and $t$ is the scaling parameter. It basically shares the same property with the standard log-barrier function, when $t$ approaches $+\infty$, our log-barrier extension would approach a hard indicator: $H(z) = 0$ if $z \leq 0$ and $+\infty$ otherwise. But its domain is not restricted to the feasible points only. This removes the demanding requirement for *explicit* Lagrangian optimization.

We are doing approximated Lagrangian optimization with *implicit* dual variables. Our strict positive gradient of $\tilde{\psi}_t$ will get higher when $z$ approaches violation and effectively push back into the feasible value set. Because our penalty does not serve as a strict barrier of the feasible set, there is possibility of overflow. However, our goal is to achieve the practical goal of fully-quantized training on low-bit hardware, as long as the majority of values stay within a high confidence interval, the approach will work in practice, as we show in experimental results (§4). Recall the scenario in Figure 1(left).

## 3.3 TAIL BOUND OF DISTRIBUTION

In this section, we demonstrate that the probability of overflowing the quantization range can be controlled for a standard ResNet model. The ResNet model has $L$ layers, each layer contains a CNN operation and ReLU as an activation. Consider at layer $l$, for each output element $O \in \mathbb{R}$ in the

layer, a CNN operation with the kernel size $k$ and channel size $c$ can be viewed as a rectified dot product between the input feature $I \in \mathbb{R}^{ck^2}$ with $w \in \mathbb{R}^{ck^2}$.

$$O = \text{ReLU}(\sum_{i=1}^{ck^2} w_i I_i) \tag{6}$$

Suppose $w_i \sim \mathcal{N}(0, \sigma_{(l)}^2)$ and $\sigma_{(l)}^2$ is dependent on the layer $l$ and the parameter $u$ we choose.

The second moment of any element $z_{(l+1)} \in \mathbb{R}$ from layer $l + 1$ can be connected with the second moment of element $z_{(l)}$ from layer $l$ as follows:

$$\mathbb{E}[z_{(l+1)}^2] = \frac{ck^2}{2} \sigma_{(l)}^2 \mathbb{E}[z_{(l)}^2] \tag{7}$$

In the He initialization, we could set $\sigma_w = \sqrt{\frac{2}{ck^2}}$ to cancel the first two terms. In this work, we want to control $\sigma_{(l)}^2$ in order to reduce the chance of overflow. We choose $\sigma_{(l)}^2$ to depend on $u$ which was defined in optimization problem (1): $\sigma_{(l)} = \sqrt{\frac{2u}{ck^2}}$. Then, we can simplify Eq. (7):

$$\mathbb{E}[z_{(l+1)}^2] = u\mathbb{E}[z_{(l)}^2] \tag{8}$$

Suppose the input feature to the first layer has second moments $E_{(1)} = \mathbb{E}[z_1^2]$, then the second moments at layer $l$ can be estimated as: $E_{(l)} = \mathbb{E}[z_{(l)}^2] = u^{l-1}E_{(1)}$. Next, for each element $z_{(l)}$ at layer $l$, its tail distribution outside the quantization region can be bounded by

$$P(|z_{(l)}| > 2) = P(z_{(l)} > 2) \le P(|z_{(l)} - \mathbb{E}[z_{(l)}]| > |2 - \mathbb{E}[z_{(l)}]|) \le \frac{\text{Var}(z_{(l)})}{(2 - \mathbb{E}[z_{(l)}])^2} \tag{9}$$

where the second inequality is established by Chebyshev's inequality, here, $z_l$ is guaranteed to be nonnegative since it's the output of the ReLU layer. We can further estimate the worst upper bound for the right hand side by considering the following optimization problem:

$$\sup_{\mathbb{E}[z_{(l)}],\text{Var}(z_{(l)})} \frac{\text{Var}(z_{(l)})}{(2 - \mathbb{E}[z_{(l)}])^2}$$
$$\text{subject to} \quad (\mathbb{E}[z_{(l)}])^2 + \text{Var}(z_{(l)}) = E_{(l)}$$

As $z_{(l)}$ is the rectified value, $\mathbb{E}[z_{(l)}] > 0$. One upper bound can be established as follows:

$$\sup_{\mathbb{E}[z_{(l)}],\text{Var}(z_{(l)})} \frac{\text{Var}(z_{(l)})}{(2 - \mathbb{E}[z_{(l)}])^2} < \frac{E_{(l)}}{(2 - \sqrt{E_{(l)}})^2} \tag{10}$$

because $\mathbb{E}[z_{(l)}] < \sqrt{E_{(l)}}$ and $\text{Var}(z_{(l)}) < E_{(l)}$. Notice that the last term can be approximated to $E_{(l)}$ when $E_{(l)}$ is small enough, and we can establish our probability bound of overflowing the quantization range:

$$P(|z_{(l)}| > 2) < \frac{E_{(l)}}{(2 - \sqrt{E_{(l)}})^2} \simeq \frac{E_{(l)}}{4} \tag{11}$$

Finally, consider the average overflow probability of $z \in \mathbb{R}$ from any layer:

$$P(|z| > 2) = \frac{1}{L}\sum_{l=1}^{L} P(z_{(l)} > 2) \simeq \frac{1}{4L}\sum_{l=1}^{L} u^{l-1}E_{(1)} \simeq \frac{E_{(1)}}{4L(1-u)} \tag{12}$$

Formula (12) is derived because in deep neural networks, $L$ is usually a large number allowing us to effectively ignore $u^L$. Therefore, the overflow probability is determined by the input's second moments $E$, the layer size $L$ and the barrier parameter $u$. In practice, we can adjust both $u$ and $L$ to control the overflowing tails. In this work, $E_{(1)}$ is set around 1.0 because the input features are normalized, and $L$ is around 50 in the ResNet-50. By choosing $u = 0.1$, the number of overflowing parameters can be controlled under 2.3%.

## 3.4 Sparsity

In order to achieve the goal of practical implementation on 8-bit hardware, we aggressively fix the range of quantization to $z \in [-2, 2]$. This leaves us with the task of mapping millions of parameters to this range. Therefore, we desire a sparse solution, and naturally, as is pointed out by Hoffer et al. (2018), we use $L^1$ penalty $|\epsilon_n|$ to encourage sparsity. Our unconstrained loss then becomes:

$$\underset{\theta}{\text{minimize}} \quad \mathcal{L}(f_\theta(I)) + \sum_{n=1}^{N} \tilde{\psi}_t(|\theta_n| - u) + \gamma \sum_{n=1}^{N} |\epsilon_n| \tag{13}$$

where $\gamma > 0$ and is tuning variable, and $\tilde{\psi}_t$ is the log-barrier extension as proposed.

## 3.5 $\mu$-law quantization codebook (MU8)

Even after sparsifying the weight, we still are left will many parameters which are non-linear distributed. Luckily, they should all be small enough by now thanks to our log-barrier constraint. Uniform quantization (e.g. INT8) would cause huge information loss in this case. Inspired by $\mu$-law's (Deng & Doroslovacki, 2006) application in audio encoding, we construct a non-linear codebook accordingly, which could be implemented on 8-bit hardware directly. As is shown in Figure 1(right), we computed all possible values of FP8(1-4-3) or FP8(1-5-2)[4]used by Sun et al. (2019), and we can see our MU8 encoding are better at handling very small values e.g. FP8 (1-4-3)'s smallest possible number is $1.9 \times 10^{-3}$, FP8 (1-5-2) is $1.5 \times 10^{-5}$, whereas our MU8 could handle $5 \times 10^{-6}$ with the cost of getting sparser in larger numbers, which we have almost eliminated using log-barrier. Let's denote our the quantization function as $Q_{\mu 8}(X)$ which take $x$ as input and output quantized value $x'$. First, following the range of $\mu$-law encoding, we set the input to Equation 15 $x = z/2$, since $z \in [-2, 2]$ in this paper.

$$F(x) = sgn(x)\frac{ln(1 + \mu|x|)}{ln(1 + \mu)} \quad -1 \le x \le 1 \tag{14}$$

where $\mu$ is a tuning variable, the larger $\mu$ is, the more non-linear it becomes, and $sgn(x)$ is the sign function. Then, we get the $y \in [-1, 1]$ as the output of $F(x)$, then we perform Stochastic Rounding step introduced by Gupta et al. (2015). e.g. $y \times (2^7 - 1) \in [15, 16]$, if $y = 15.5/(2^7 - 1)$, then $P(\hat{y} = 16/(2^7 - 1)) = P(\hat{y} = 15/(2^7 - 1)) = 0.5$; if $y = 15.1/(2^7 - 1)$, then $P((\hat{y} = 15/(2^7 - 1)) = 0.9, P(\hat{y} = 16/(2^7 - 1)) = 0.1$, as shown in the following equation.

$$\hat{y} \leftarrow \text{Stochastic Rounding}(y, LL, UL) = \begin{cases} LL & \text{w.p. } 1 - (y - LL) \\ UL & \text{w.p. } (y - LL) \end{cases} \tag{15}$$

where $UL$ indicates upper limit and $LL$ indicates lower limit. At last, we get the quantized value $x'$ via the inverse function of encoding function (15):

$$x' \leftarrow F^{-1}(y) = sgn(y)\frac{1}{\mu}((1 + \mu)^{|y|} - 1) \tag{16}$$

## 3.6 Other Useful Techniques

**Straight-through Estimator (STE)**: We adopt the same STE as Zhou et al. (2016) which is critical to the convergence of our models. The gradient of the quantization steps in the quantization model is based on the straight through estimator.

**Chunked Updates**: We also use the same gradient accumulation strategy as Sakr et al. (2019); Wang et al. (2018). During updates, since we have quantized the gradient and weight parameters using $Q_{\mu 8}(X)$, the gradient will be lower bounded by the smallest possible value of the MU8 encoding($10^{-6}$), which is small enough to preserve model accuracy within the 8-bit constraint. However, directly adding this small gradient to a large number (weight) in our MU8 quantization would cause this small gradient to be rounded-off and thus lose information due to the growing

---

[4]FP8(1-4-3)'s largest value is [-240, 240], and FP8(1-5-2)'s is [-57344, 57344], they are better at handling long-tails than MU8, but we are not interested in large numbers in the long-tail.

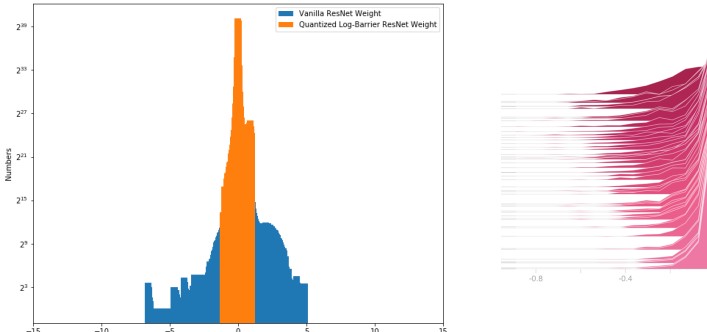

Figure 3: Left: LogBTQ ResNet-50 weight distribution with and without log-barrier loss throughout the training process. Right: Example weight distribution in a conv layer of LogBTQ ResNet-50.

intervals of MU8 encoding at larger values. Performing chunked gradient updates, updating the weights only after $k$ steps of gradient descent (in this work, we set $k = 20$), helps to accumulate the gradients to be large enough to be rounded up, aiding convergence.

## 4 EXPERIMENTAL RESULTS

Table. 2 shows the performance of models trained with LogBTQ from scratch on the MNIST, CIFAR-10, and ImageNet datasets. Our performance on CIFAR-10 is even higher than the FP32 baseline, and our results using ResNet-50 trained from scratch on ImageNet are competitive against both of the works in comparison. This is particularly impressive given that LogBTQ uses strictly 8-bits to store gradients whereas previous work resorted to FP16 or hybrid 8/16. Our accuracy loss of ResNet-50 on ImageNet is only 2.48% which for many use cases is worth the benefit of keeping everything in only 8 bits of precision. For practical purposes, 2-5% accuracy loss should be tolerable if calculations can be kept strictly in 8 bits.

| Training Scheme | MNIST ResNet-18 | | CIFAR-10 ResNet-18 | | ImageNet | | | |
| | | | | | ResNet-50 | | MobileNet | |
| | FP32 | 8-bit | FP32 | 8-bit | FP32 | 8-bit | FP32 | 8-bit |
|---|---|---|---|---|---|---|---|---|
| Wang et al. (2018) | — | — | 92.77 | 92.21 | 72.14 | 71.72 | — | — |
| HFP8 (Sun et al., 2019) | — | — | — | — | 76.44 | 76.22 | 71.81 | 71.61 |
| LogBTQ FP8(1-4-3) | 99.9 | 98.1 | 94.08 | 92.31 | 73.59 | 54.32 | 71.68 | 50.19 |
| LogBTQ FP8(1-5-2) | 99.9 | 98.3 | 94.08 | 93.21 | 73.59 | 58.46 | 71.68 | 51.33 |
| **LogBTQ MU8(ours)** | 99.9 | 99.6 | 94.08 | **94.50** | 73.59 | 71.11 | 71.68 | 68.41 |

Table 2: Results of LogBTQ training compared with the two most relevant low-precision training schemes. Note that the previous work listed here is *not* directly comparable to ours; as is shown in Table 1. Though these works perform the majority of computation in 8 bits, they still have the advantage of FP16 or Hybrid FP8/16 when accumulating gradients, and dynamic quantization range. First 2 rows are numbers reported by Wang et al. (2018) and Sun et al. (2019).

## 5 DISCUSSION

Hou et al. (2019) also provides a bound for the quantized gradient of QAT models, giving the intuition that quantized gradients would slow down convergence, which we also observe in our training.

For each arithmetic operation, we perform our $Q_{\mu 8}(X)$ quantization right after each operation to ensure the number still falls into 8 bits. For out of range numbers, we clamp. We verify our assump-

tions in our experiments: Figure 3 (left) shows a layer's weight distribution during training, and we can see that they are nicely bounded by our log-barrier to within [-1.5, 1.5]. Figure 3 (right) shows the weight distribution is concentrating to smaller values during training.

Since we only have $10^{-6}$ precision to handle the gradients, we are inevitably losing some accuracy compared with higher precision training schemes. FP16 can handle $10^{-8}$ precision and FP32 can handle $10^{-32}$ precision. As we can see in Table 2, LogBTQ can perform almost perfectly on easier tasks such as MNIST and CIFAR-10, but degrades when we are training on the more challenging ImageNet dataset. Our Mu8 encoding performs better than 1-5-2 and 1-4-3 with our LogBTQ quantization scheme, showing $\mu$-law encoding can handle small fixed quantization range better than 1-5-2 or 1-4-3. As is pointed out by Sheng et al. (2018), MobileNet architecture has some layers that are unfriendly for quantization, e.g. An outlier in one channel could cause a huge quantization loss for the entire model due to an enlarged data range. We employed the same techniques proposed by Sheng et al. (2018), though MobileNet benchmark is still not as good as ResNet-50 due to the agressive reduction of parameters.

Overall, it still depends on the use case of the deep learning models to consider the trade-off between accuracy and precision to choose which quantization scheme to adopt. As far as we are aware, LogBTQ is the first full 8-bit quantization scheme with competitive performance on a large-scale dataset such as ImageNet. This opens up the prospect of training fully-quantized models on low-precision hardware.

## 6 CONCLUSION

Motivated by the limitations of *fake* quantization, we propose Log Barrier Tail-bounded Quantization (LogBTQ) which introduces a log-barrier extension loss term that enforces soft constraints on the range of values that model parameters can take on at every operation. Our approach eliminates overflow in practice, sparsifying the weights and using $\mu$-law non-uniform quantization, allowing for fully quantized 8-bit training of deep neural network models. By constraining the neural network parameters driven by theoretical motivations, this work enables the possibility for the first time fully-quantized training on low-precision hardware.

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
