# OpenReview forum: "End-to-end Quantized Training via Log-Barrier Extensions"
_ICLR.cc/2021/Conference — Reject_

### Official Review · AnonReviewer4 · 2020-10-26
**Official Blind Review #4**

**Rating:** 3
**Confidence:** 5

**Review:**

The authors introduce a log-barrier extension loss term enforcing soft constraints on the range of values to enable fully end-to-end quantization-aware training.

Strengths of the paper:

- The paper addresses an important topic, because there are increasing concerns in performing fully end-to-end low precision training to deploy on low-precision hardware.
- The method has a practical goal and could be interesting for practitioners

Weaknesses of the paper:

- Lack of positioning with respect to the SOTA quantization-aware training(QAT) and post-training quantization(PTQ) schemes, there are plenty of missing related literatures on both quantization schemes. Some statements in the background and related work could be wrong. For example, QAT also focuses on efficient inference as well as PTQ. The levels of practical applicability of a variety of quantization solutions have been introduced in DFQ(Nagel etal., 2019). Survey on the related work is not sufficient.
- Having a benchmark would be interesting if it will include some SOTA methods and evaluates with them. The comparison targets are mostly out-of-date. It is lack of convincing evaluation results to support the proposed scheme.
- Organizing the whole contents is ok but not good enough for the readers to easily follow and understand.

Detailed comments:

(1) The terminology on swamping might not be familiar with the ML community. Explaining the criticality of swamping problem is not good enough in the intro. You should provide how critical the problem is on the low-precision hardware with the other SOTA quantization schemes. For example, the probability of occuring swamping without applying the proposed scheme, etc.

(2) Evaluation results provided in the paper are just for comparing accuracy. Accuracy loss is intrinsic in fully end-to-end low-precision training. The benefits of employing the proposed scheme would be beyond accuracy, say memory or energy-saving constraints for on-device training. Experimental evaluations to support the necessity and merits of the proposed scheme should be provided.

(3) The quantization range is fixed in the proposed scheme. Is it a merit that the proposed scheme does not need to adjust the range and precision either per-layer or per-channel during training as in other SOTA methods?

(4) Writing on the constrained optimization formulation is a bit verbose and not properly formulated.

(5) Inducing the tail bound of distribution to demonstrate that the probability of swamping can be controlled, several assumptions and approximations have been applied for the worst-case upper bound. Are the assumptions reasonable to work in practice? For example, assuming that the weight distribution is Gaussian is too strong to be practical.

(6) The paper has a conceptual overlap with other quantization approaches and some of the proposed scheme is not entirely novel resulting in a weak contribution.

(7) In Table 2, the MobilNet has more severe degradation of accuracy than the ResNet on the low-precision(8-bit) setting. Could you explain why this happens?

Minors:
- Several typos: There was been in p.2, to soft threshold the range of in p.3, theta-i in eq.(3), some more in p.7

---

> ### Author Response · Authors · 2020-11-13
> **QAT References not directly comparable**
>
> Thank you for your review. **We hope R4 could reevaluate our work based on our revision!**
> - Actually, our approach is **not directly comparable** to existing QAT.  They focus on dynamic range training to maximize accuracy, but this means that their approaches lose the ability to actually train on fully 8-bit hardware. For example, their operations require 16-bits for summation or different floating-point encodings for the forward and backward pass. In contrast, our approach is designed to enable end-to-end training on 8-bit hardware.
> - We will try to make this more clear in our revision and add more QAT comparison, but we position our paper with respect to all the most relevant works that we know of, as recent as papers published in 2020 Could you suggest the literature you have in mind that we have missed besides [Sakr et al. 2019] and [Zhang et al. 2020]? (Those works are relevant but also not directly comparable -- see our response to R2&R3 who pointed them out.) (**We have updated our related work section in the revision**)
>
> (1) We will certainly fix the confusion around swamping and provide more detailed background in our revision. (**fixed in our revision**)
>
> (2) Accuracy for other QAT models involving FP16 or FP32 in its training steps are **not comparable** to our work since they involve fake quantization. If we allow FP8 per layer dynamic range fake quantization, our result can be as accurate as FP32. Our methods do not need higher precision and would make it perfect for low-bit on-device training. We will add the merits of our methods in terms of energy-saving in our revision.
>
> (3) Yes, this is a huge improvement since we can actually ease the hardware implementation significantly. Dynamic range training cannot be trained on standard 8-bit hardware.
>
> (4) Since our work spans a broad background ranging from hardware design to optimization, it is necessary to give more detailed motivation to present to the readers. Could you specify how the optimization is not properly formulated so that we can improve it in our revision?
>
> (5) It is reasonable since this matches with our empirical results: the tail bound suggested cut-off range works well in these datasets. Also, ResNet itself was motivated by a similar bound. **We have discussed this in our revision, and that is the same as what pointed out by [Sakr et al. 2019]**
>
> (6) We are not sure which work we are overlapping with. Could you please specify which previous work you are thinking of? We plead R4 to re-evaluate our work in light of our response and the other reviews.
>
> (7)As for MobileNet results, we are still investigating the results and will provide more detailed analysis in our revision. **We have updated our mobileNet results using the techniques mentioned in [Sheng et al. 2018: Aquantization-friendly  separable  convolution  for  mobilenets].**
>
> We have fixed all the typos in our revision.

---

### Official Review · AnonReviewer2 · 2020-10-27
**Log barrier extensions to improve QAT**

**Rating:** 5
**Confidence:** 5

**Review:**

The paper presents a set of well crafted regularization techniques that improve the accuracy of QAT.

The overview of related works, particularly QAT is very well done. We clearly see what the present limitations are. I would like to point out that there have been works that have analytically determined suitable fixed quantization ranges in the context of backprop. The authors claim otherwise, I would suggest including works such as [1] in section 2.2, and revisit that claim.

The proposed method method is very nice, Lagrangian optimization techniques are used to ensure weights do not fall into the tails of the distribution. This reduces the probability of overflows.

I wish to point out that the term 'swamping' does not describe an overflow phenomenon per se, it is rather an underflow that occurs when adding two very different floating-point numbers. Avoiding tail samples is definitely useful to reduce swamping, but some of the wording in section 3.3 and in the introduction probably ought to be revisited. It is not a very serious issue, but the clarity in that regard could be improved. Otherwise, the analysis in Section 3 is excellent.

One question about eq. (9): the tail distribution is defined as P(z>2) (subscripts aside for convenience). Shouldn't it be P(|z|>2)? Won't large negative numbers be problematic as well? In the grand scheme of things this won't affect the conclusions, but I think this correction (or at least a comment) should be made for extra rigor.

The experimental results are good. I wonder if the authors could spare a few sentence to discuss how they simulated accumulation quantization. This issue is not as straightforward as representation quantization that would apply to activations/weights/gradients. The cited papers that have looked at swamping previously (Wang & Sakr) do mention something about modifying the deep learning framework at the GEMM level. Is the same technique used?

One question about a statement made in the conclusion: Throughout the paper, a clipping level of 2 has been chose, but in the final discussion, it is stated that the weights are contrained to the range [-1.5,2.5]. Why the disparity? Is this an error or something expected? If so, can it be explained?

Minor issue:
It is stated in section 3 that \phi_t approaches the indicator function when t approaches infinity. How so? I get that z<0 -> \phi_\ifty = 0 but z>0 is not well defined. Maybe the definition of \phi for z>0 can be explicitly stated.

[1] Sakr, C., & Shanbhag, N. Per-tensor fixed-point quantization of the back-propagation algorithm. In 7th International Conference on Learning Representations, ICLR 2019.


=======================================================================================================
Comments Post Rebuttal:
I still find the technical contribution of this paper to be a good one. However, As stated in my original review, there were some clarity issues with the manuscript. While, I personally was able to follow along, the other reviewers are correct in their claim that the writing can be quite confusing (Reviewer 4 makes a strong case). To that end, I have decided to decrease my score to a 5 because I can no longer say the manuscript is ready for publication. Nonetheless, I urge the authors not to be disappointed. The presented work has many merits, and I am sure that with a thorough "clean up" of the text, this paper can be a great one!

---

> ### Author Response · Authors · 2020-11-13
> **Please give us an opportunity and reevaluate based on our revision!**
>
> We appreciate R2’s encouraging words and insightful feedback but we hope R2 **could continue supporting us based on our newly finished revision!**
> - We will cite [Sakr et al. ICLR 2019] and compare their findings with ours. [Sakr et al. ICLR 2019] first defines a framework based on analyzing the representational, computation and communication cost of different network layers and determines what bit precision they would need accordingly, which results in mixed precisions. Whereas we decide what bit-width we are allowed on hardware to begin with, and discuss how we can make training on fixed low-bit-precision hardware feasible. We will follow your suggestion to compare with their framework and provide an analysis in our revision.**We have included this in our revision:** Regarding [1] Sakr et al. They defined predetermined dynamic range (PDR), and mentioned in **their appendix B**:” Because FX numbers require a constant PDR, clipping of gradients is needed since their dynamic range is arbitrary. Ideally, **a very small PDR would be preferred** in order to obtain quantization steps of small magnitude, and hence less quantization noise. “ This is exactly the same intuition that we have, and we managed to achieve **fixed PDR with log barrier**. Their clipping rate definition also are equivalent to our tail-bound definition.
> - Sorry about the misuse of swamping confused with overflowing. We are fixing that in our revision. We will add notes to specifically distinguish the difference between swamping and overflowing.**(we have fixed this issue in our revision)**
> - Thank you for pointing out our analysis of tail distribution! Yes, it should be $P(|z|>2)$. We will make sure we fix this issue in our revision. **We have fixed this issue in our revision:** About $P(|z|>2)$
> In general, we should consider $P(|z|>2)$ to prevent overflow issue, however, in our case where the input for each layer has been rectified by the previous ReLU layer, so $z$ is guaranteed to be nonnegative, therefore, P(|z|>2) = P(z>2)
>
>
> - Regarding simulated accumulation quantization: Yes, we used the same techniques as Wang & Sakr which involves gradient accumulation and chunk updates as well as stochastic updates. We will provide more details in our revision.**(we have fixed this issue in our revision)**
> - Disparity in range [-2,2] vs [-1.5,-1.5]. This is a minor disparity, we did set the quantization range to be [-2, 2], but our empirical result showed most values lie between [-1.5,1.5] in practice on these data.
> - About log barrier extension, as we cited [Kervadec et al. 2019] in the paper, the extension only handles the corner case, while the log-barrier still behaves like the original log-barrier which mimics an indicator function. When t approaches $+\infty$, our log-barrier extension would approach a hard indicator: $H(z) = 0$ if $z \leq 0$ and $+\infty$ otherwise. Thanks again for your detailed comments, and we will adjust our descriptions in our revision to explicitly state the domain of $\phi$.**(we have fixed this issue in our revision)**

---

> ### Author Response · Authors · 2020-11-25
> **Please support us!**
>
> Dear R2:
>
> Thank you again for your encouraging words! Unfortunately, it seems like we are not getting much discussion going on here before the rebuttal phase ends.
>
> Since you were the only one to offer feedback during the discussion period, we are here to make a plea for support based on our revision. As we have tried to address all the concerns regarding the clarity issues in the initial feedback, we believe the current version of the manuscript should be ready for publication. Our work contains novel ideas and presents practical low-bit quantization methods which are well-motivated by theory. We tried our best to showcase its performance and added more results in the revision, and we will release our implementation if we could be given the opportunity. We believe our paper is not a cookie-cutter variant of yet another quantization paper, and it could be of value to the quantization community at ICLR!
>
> Meanwhile, we also realize that our work spans a broad knowledge background ranging from hardware design to optimization, and it could result in confusions as happened in the initial version. It also could cause readers from diverse ML background to be less familiar with the niche background/literature of end to end low-bit training (w/o mixed precision) as we covered in *Table 1*.  Therefore, we understand and respect that even ML experts could form a different view of our presentation.
>
> We implore you to support us given your great expertise about the topic!

---

### Official Review · AnonReviewer3 · 2020-10-28
**The paper is addressing an important and challenging problem of end-to-end training of deep nets in fixed-point, in this case, with 8-bit precision. The results promising but sparse. The work needs additional results/comparisons before it becomes ready for prime time.**

**Rating:** 6
**Confidence:** 5

**Review:**

The paper is addressing an important and challenging problem of end-to-end training of deep nets in fixed-point, in this case, with 8-bit precision. A good solution to this problem can have a major impact on the deployability of deep nets on embedded hardware. The basic idea is to introduce an additional term (the log-barrier constraint) in the loss function to constrain the allowable range over which model parameters are allowed to take values. The authors use of mu-encoding to assign non-uniform quantization levels to minimize the quantization error. The main results are in Table 2 showing that the method eliminates overflow in practice and allows quantized networks to approach the accuracy of full-precision networks on the MNIST, CIFAR-10 and ImageNet.

A few comments/questions:

1) How was the quantization range ([-u,+u]) chosen? There will be a trade-off between the precision (8-b vs. 6-b) and the quantization range u. Is there any notion of optimality of the chosen value of u for 8-b quantization.
2) How important is mu-law coding? How much accuracy is lost when uniform coding within the quantization range is applied? Please note, doing arithmetic on non-uniformly quantized variables is harder than with uniform coding, which has a bearing on hardware realizations. It will be good to quantify the accuracy loss in the absence of mu-law quantization.
3) The results in Table 2 are sparse. It further indicates that the accuracy of MobileNet using the proposed method is quite significantly lower (66% vs. 72% for Sun) compared to previous works. MobileNet is designed for embedded IoT type hardware. 8-b or even 9-b quantized MobileNet results would put this work firmly in the domain of end-to-end fixed-point training of lightweight networks and hence make it more useful. These results will strengthen Table 2 significantly.
4) Please comment on the choice of hyperparameters and initialization. Is this method robust to these choices?
5) How does 8-b fixed-point compare with say MiniFloat-8 or equivalent? Such low-precision floating-point realizations are an attractive choice and a competitor to fixed-point.
6)The paper is missing comparisons with a couple of highly relevant papers on fixed-point training listed below:

[1] Zhang et al., Fixed-point Back Propagation Training, CVPR 2020.
[2] Sakr et al., Per Tensor Fixed-Point Quantization of the Back Propagation Algorithm, ICLR 2019.

Overall a very nice paper but needs more work (indicated above) to strengthen the results.

---

> ### Author Response · Authors · 2020-11-13
> **Mobilnet-experiment included in the Revision**
>
> We thank R3 for your constructive feedback!
>  1. Quantization range $[-z, z]$ was chosen empirically in this work. We acquired the statistics of the weight vector of a pre-trained FP32 full-precision model and determined our desired quantization range, it could also be determined by hyperparameter search. Another important factor is the tail bound from our derivation. We select the $z$ value by using the $P(x>z) < 0.05$. Also, in our experiment, we do observe the majority of the values (weight, gradient, loss, activation and etc) falls in the range. Note that unlike approaches with dynamic range, we only need to find one value.
>  2. Mu-law encoding is quite crucial in our case. As we pointed out in Figure 1 right, Mu-law encoding has higher resolution compared to FP8 (1-5-2) and FP8 (1-4-3). If we directly use the uniform encoding between -2 to 2, which have a resolution of $\frac{4}{256} = 0.0156$. Then the network on ImageNet learns nothing, i.e. $acc = 0.001$. We will also test the uniform 1-5-2 only (or 1-4-3 only) for forward, backward, every arithmetic operation for comparison.
>  3. We have updated our results for MobileNet in our **revision using the techniques mentioned in [Sheng et al. 2018: Aquantization-friendly separable convolution for mobilenets]**, and this turned out to be where our previous MobileNet implementation lost accuracy.
>  4. Yes, the methods are robust to random initialization. We will make our implementation **public on GitHub** once the paper is accepted for publication.
>  5.  Yes, tested the uniform MiniFloat with 1-5-2 only (or 1-4-3 only) and compared with Mu law, and we have included our comparison in the revision. The reason that we select mu-law is that by using mu-law, we can control the density of numbers within a range.
>  6. We will cited and compared with these newly published works. [1] Zhang et al., Fixed-point Back Propagation Training, CVPR 2020. [2] Sakr et al., Per Tensor Fixed-Point Quantization of the Back Propagation Algorithm, ICLR 2019. It seems both methods still involve dynamic range per layer and dynamic bits in forward and backward propagation. Thus, they are still not directly comparable to our work. We have included them in our **revision**.

---

### Official Review · AnonReviewer1 · 2020-10-29
**Not sure if this paper is technically sound**

**Rating:** 3
**Confidence:** 4

**Review:**


This paper introduced a log-barrier based regularization method to reduce the dynamic range of data types (activation, weight, error, gradient, and input) in neural networks. The authors claim that such regularization is important to avoid overflow or swamping in the accumulation of matrix multiplication.

However, the reviewer is afraid to think that there are serious technical issues with this claim. Here are a few details regarding this concern.

- The main claim of this paper seems to be that by reducing the dynamic range of the neural network data, they can avoid overflow. However, it does not seem to be (always) true; what matters more would be the resolution of information you need (or want) to keep, and the number of data elements you accumulate. E.g., assume that we use 8-bit -- even if the magnitude of data is constrained to be around 2^-3, if the resolution of info is 2^-8, there will be an overflow after 8 times of accumulation. Also, note that the magnitude of activation or weight of a layer is often not important due to scale-invariance of the normalization techniques (like batchnorm). Thus it is not clear if constraining the magnitude of data is always beneficial.

- The authors claim that previous reduced-precision training techniques (like [Sun et al., NeurIPS19]) have dynamic quantization range across the layers. This is NOT true, since the reduced-precision floating point format has the fixed dynamic range; e.g., [Sun et al., NeurIPS] employs (1-5-2) format for representing back-prop error, implying the fixed dynamic range of 2^(-15) -- 2^(16). In fact, one very compelling benefit of using the reduced-precision floating-point over the fixed-point format is that the floating-point format does not need scaling of the magnitude across the layers.

- It is not clear if the authors' understanding of "overflow" or "swamping" is the same as what is reported in prior work (e.g., [Sun et al., NeurIPS19]).  First of all, overflow and swamping issues are different - the former issue is for the fixed-point accumulation where the accumulated value wraps around to the small value when it exceeds the largest value representable by the given accumulation precision. Whereas, the latter is for the accumulation of floating-point numbers, where the small magnitude value is ignored (or truncated) when it is added to the large magnitude sum. Thus, it is very confusing when the authors use these two different terms interchangeably in the text (e.g., the 3rd paragraph of Intro).

- There are many confusing points in the analysis of the overflow in Sec 3. First of all, the authors seem to claim that Prob(z > 2) is the overflow condition... but why? I couldn't find the notion of bit-width in this analysis... but then how can we check the overflow?

- The authors mention that they employ STE, which does not make much sense in case of the quantized training; what does it mean to take STE for the quantization of back-prop error?

- It is not clear how the proposed method is evaluated. The computational graph in Fig 2 does not seem to include the accumulation quantization. Also, more detailed information should be provided about the following; how the MU8 format is implemented to use 8-bit hardware? how the proposed method is implemented in the deep learning framework?


To sum up, this paper seems to have several technical flaws, which should be carefully addressed.

---

> ### Author Response · Authors · 2020-11-13
> **We believe we are technically sound and we fixed our wording in the Revision.**
>
> Thank you for your review.
> - This is exactly right. We apologize that we might have confused overflowing with swamping in the text, and we are editing to fix this mistake. Regardless, we are trying to tell the same story that we need to have high resolution (e.g. via our proposed mu-law) when we constrain the network values within a small range using the log barrier constraint. Log-barrier incurs a huge penalty as the value approaches the threshold. We theoretically and empirically show the probability of the overflow number in the neural network is less than 5%. **(We have fixed this issue in our revision)**
> - We agree with the reviewer’s statement about [Sun et al. 2019] that they do not have layer-wise dynamic ranges. However, [Sun et al. 2019]’s forward is 1-4-3, the backward is 1-5-2, and thus, the resolutions are different. Even though you use the fixed-point resolution for backward or forward, the system has to keep using two types of precision for weight and gradient, this would cause rounding issues. Also, as we show in the density of values using our mu-law approach, by adjusting the mu-law encoding, the resolution could go as low as $10^{-16}$, which is far more refined than the suggested $2^{-8}$.
> - Thank you for pointing this out! We will clarify in the revision. We use a log-barrier constraint to get around the first issue: keep the fixed-range small enough. And chunk update, mu-law, stochastic rounding to get around the second issue: swamping in this case. **(We have fixed this issue in our revision)**
> - This is not related to bit-width operation. This shows the percentage of values in the network larger than 2, which shows the effectiveness of log barrier. The bit-width is in the mu-law section. Mu-law is one of the fundamental encoding rules.
> - Again, we are using the mu-law for quantization. There is a hard quantization step. Thus, in the backprop step we use STE to pass gradients if the step involves hard quantization. STE has been used to enable QAT in a number of previous works such as Dorefa-Net[Zhou et al. 2016] and others. In ICLR 2019 there was this work specifically discussing this: “Understanding Straight-Through Estimator in Training Activation Quantized Neural Nets” [Yin et al. 2019].
> - Mu-law is one of the most widely used encoding rules in the audio domain, but the idea could be easily adapted to quantization. In the current PyTorch implementation, we quantize all values after each single arithmetic operation. However, in a real hardware situation, this could be directly solved by mapping these values on the input and output.
>
> Hope our clarifications could address your concerns.

---

### Author Response · Authors · 2020-11-23
**We have updated a revised version of the paper!**

Dear reviewers and ACs:

We have updated a revised version of the paper taking all the reviewers comments into consideration.
**Please kindly take a look and re-evaluate** our work based on our **revision**.
- **@R2**: We noticed you adjusted your score and comments this morning before we uploaded the revision, could you kindly take another look?

We have made the following changes in our **revision**:
1. Fixed confusion between **“swamping”** and **“overflowing”**
2. Clarified $P(|z|>2)$ needs to be *absolute values*
3. **Rerun MobileNet experiments** and utilized techniques proposed by *[Sheng et al. 2018: A quantization-friendly separable convolution for mobilenets]*. We bumped our MobileNet result to **68.4%** accuracy.
4. We added more details about *stochastic rounding, chunk update, and mu-law*.
6. We added more detailed description about log-barrier extension
7. We compared with [Sakr et al. 2019] and [Zhang et al. 2020] and added more discussion about how they share the **same intuition** but they are different since they used *dynamic quantization range*.
8. We ran additional **FP8(1-4-3)**, **FP8(1-5-2)** experiments in our LogBTQ quantization pipeline and compared with our **MU8** encoding scheme.
9. We fixed the typos accordingly.
For each of your individual questions, we have **updated our responses to each question** in the comments below.

---

### Decision · Program_Chairs · 2021-01-07
**Final Decision**

**Decision:**

Reject

**Comment:**

This paper introduced a log-barrier based regularization method to reduce the dynamic range of data types in neural networks. As pointed out by the reviewers, there are many technical issues. The authors agree with the reviewers in the rebuttal, though claimed that they are fixed in the revised version of the paper.

Experimental results are not convincing. It is not clear how the proposed method is evaluated. Accuracy of MobileNet using the proposed method is quite significantly lower compared to previous works. The work needs additional results/comparisons with other highly relevant papers on fixed-point training.

There are also many clarity issues that need to be fixed.